# A preclinical model of post-surgery secondary bone healing for subtrochanteric femoral fracture based on fuzzy interpretations

Pratik Nag[1], Souptick Chanda [1,2]*

**1** Department of Biosciences and Bioengineering, Indian Institute of Technology Guwahati, Guwahati, Assam, India, **2** Mehta Family School of Data Science and Artificial Intelligence, Indian Institute of Technology Guwahati, Guwahati, Assam, India

* csouptick@iitg.ac.in

**Data Availability Statement:** All relevant data are within the paper and its Supporting Information files.

## Abstract

Mechanobiology plays an essential role in secondary bone fracture healing. While the introduction of newer type of plates, e.g. locking plate (LP), is becoming increasingly popular for complex femoral fractures, the conventional technique involving dynamic compression plate (DCP) remains the standard choice. The difference between the two techniques lies primarily in their screw fixation mechanisms. The present study applied 3D dynamic fracture healing scheme modelled on a subtrochanteric femur fracture, regulated by both finite element (FE) analysis and Fuzzy logic control in order to understand the spatio-temporal healing phenomena for both LP and DCP. The study further examined the influence of the two screw fixation mechanisms in determining the comparative progression of fracture healing. The problem was solved iteratively in several healing steps running in loop and accordingly, the local tissue concentrations and material properties were updated. The predicted results accorded well with various previous experimental observations. The study found an initial delay in healing associated with DCP. However, as the healing progressed, there was no significant difference in overall callus modulus. The presented preclinical model may further help predict bone healing for different implantation techniques, and thus can serve as a non-invasive tool for evaluating relative merits of extramedullary plating techniques.

## 1. Introduction

Incidences of femoral fracture can be the outcome of traumatic (e.g., traffic accidents or fall etc.) or non-traumatic conditions (e.g., osteoporosis, geriatric population, cancer in bone etc.) [1]. Biological fixations are used to bridge the fracture site and impart stability to promote bone healing. Primary bone healing involves direct union of two broken bones, often seen in case of very small fracture gap (10–100 μm) and strain below 2% [2]. On the contrary, secondary bone healing is the most prevalent one and involves classical stages of healing i.e., injury, haemorrhage, inflammation, primary soft callus formation, callus mineralization and callus remodelling. This type of healing is primarily observed in fracture gaps with larger micromotion.

**Funding:** SC SRG/2019/000235 Science and Engineering Research Board https://serbonline.in/SERB/HomePage NO.

**Competing interests:** The authors have declared that no competing interests exist.

Major proportion of the femoral fractures are treated with internal plate fixations that differs primarily in the screw design. Conventional plating technique is based on the fixation of dynamic compression plate (DCP) [3]. DCP involves the use of compression screws that relies on preloading of several interfaces in the construct [4]. The tightening of the screws results in generation of force at the screw thread-bone interface, and predominantly between the bone and plate [5]. The introduction of newer type of plate, locking plate (LP), has expanded the ways of screw-plate fixation. LP involves locking screws having threaded head that allow the screw to fasten into the plate as well as into the bone [6]. The biomechanical environment achieved using LP is totally different from that obtained through DCP [6]. The surrounding environment of the screws in DCP is governed by a mix of effects contrary to LP, whose mechanics is simple like that of unilateral external fixation device. The bony fragments are not compressed and the fracture surfaces exhibits small amount of elastic motion. As compared to DCP, LP are less rigid and forms fixed angle between the plates, screws and bone fragments. The fixed angle eliminates the requirement for compression or contact between the plate and bone. This suggests that the plate can be offset from the bone, thereby preserving the periosteum and avoiding necrosis. Various authors have observed that LP performs better than DCP in older or weaker bone [3,7,8]. Few other biomechanical studies found LP to have more stiffness and superior failure load compared to other implants [9–11]. Uhl et al. [12] found that DCP provides inherently more stability than LP. Gap closure and bone-to-bone load transfer resulted in better mechanical performance of DCP over LP during axial compression [12].

A number of numerical models of the bone fracture healing have developed over the last few decades to predict quantitively tissue differentiation. Those models can be broadly classified as mechanoregulatory healing model, bioregulatory healing models, and coupled mechano-bio-regulatory healing models. In mechanoregulatory models, the mechanical stimulus regulated the tissue differentiation in the callus region. Claes et al. [13] and Claes and Hiegele [14] correlated strains and hydrostatic pressure to callus tissue differentiation using elastic FE models. Few other studies also used FE models to predict stress and strain distribution at particular healing stages of the fracture callus region [15–17]. Their result suggested hydrostatic stresses have an effect on revascularisation and tissue differentiation. Various studies have adopted dynamic models to simulate the healing as a time dependent feedback regulation system [18–25]. Most of these studies used strain invariants as mechanical signals to predict tissue differentiation [19–21,24] while a few others, in addition used fluid velocity [20,22] and fluid shear stress [25]. On the other hand, bioregulatory and coupled mechano-bio-regulatory healing models include activities (migration, proliferation, differentiation and death) of different cells participating in the callus region. Few of these models included growth factors to regulate tissue differentiation [26,27] while others used concentration of different cell types [20,22,23,26]. Fuzzy logic-based algorithms have successfully been employed in determining tissue differentiations. Amnet and Hofer [18] simulated kinetics of the healing process using linear elastic FE simulation in combination with their fuzzy logic model. Simon et al. used fuzzy logic-based algorithm to predict diaphyseal fracture healing [28]. Also, Shefelbine et al. [29] used fuzzy approach in their study to determine trabecular fracture healing. Later in another work, Simon et al. incorporated blood perfusion as a spatio-temporal state variable to simulate revascularisation process [30]. Whener et al. used the fuzzy logic algorithm developed by Simon and Shefelbine [25], to study the influence of fixation stability on the healing time.

Several experimental [7,12,31,32] and computational studies [33–36] have compared and evaluated locking and compression screws based on their mechanical performances. To the best of the authors' knowledge, no modelling approach addressed the influence of screw type in determining the overall progression of fracture healing. Furthermore, most of these investigations relied on 3D primitive models instead of realistic bone and implant geometries. The

objective of the present study, therefore, was to apply a 3D dynamic fracture healing scheme modelled on a subtrochanteric femur fracture, regulated by both FE analysis and Fuzzy logic control in order to understand the spatio-temporal healing phenomena for both LP and DCP. The study is based on a hypothesis that fuzzy logic controls could actually be used to predict tissue transformation in a realistic and non-invasive manner for secondary healing in plated bone. We further hypothesized that the screw fixation mechanisms herein bear a significant influence on the healing outcome.

## 2. Materials and methods

The simulation of the healing phenomenon in the callus region can be described as an initial value problem involving two mechanical (invariants of strain tensor) and five biological state variables (local tissue composition and vascularisation). The mechanical stimuli were firstly calculated using the finite element analysis (ANSYS) after properly stating the initial condition in the pre-processor along with loading and boundary conditions. All seven state variables were taken as input to predict the tissue differentiation and vascularisation in the bone healing region using linguistic rule based fuzzy logics (MATLAB) with 32 single rules. Accordingly, the local tissue concentration and material properties were updated. The problem was solved iteratively in several healing step (Fig 1) running in loop.

### 2.1 Mechanical part: Calculation of mechanical stimuli

For both compression plating and locked plating, the 3D FE models of femur were generated using the manufacturer supplied CAD model of the left femur (Sawbones Europe AB, Malmo, Sweden, model# 3406). A 20mm fracture gap in the subtrochanteric region of the CAD femur was simulated followed by virtual implantation in the NURBS modelling environment of Rhinoceros v14.0 (Rhinoceros, Robert McNeel & Associates, Seattle, USA) (Fig 2). Thereafter the callus region representing the initial fracture haematoma, was modelled in the implanted femur to capture the tissue composition resulting from biological tissue characterization. This potential healing region was shaped as an ellipsoid around the fracture site [37] (Fig 2A). The tessellated surface models were then imported into Ansys ICEM CFD v19.0 (ANSYS Inc., PA, USA) to generate volumetric meshing comprising of 4-noded tetrahedral elements (Fig 2C). After that the volumetric meshing was imported into Ansys mechanical v19.0 (ANSYS Inc., PA, USA) for further analysis. A mesh convergence analysis was carried out with three different mesh size of 0.5 mm, 1.0 mm, and 1.5 mm constituting 427113, 303869 and 262362 elements respectively in the callus region. For mesh sizes 0.5 mm, 1.0 mm and 1.5 mm, the values of distortional strain were predicted to be 0.1309, 0.1311 and 0.1050. Thus, doubling the mesh size from 0.5 mm (fine sized mesh) to 1.0 mm (medium sized mesh) in the callus region resulted in change of the distortional strain by 0.1% whereas the change was almost 20% from medium sized mesh to coarse sized mesh (1.5 mm) Similarly, the variation of the peak von Mises stress between fine and medium mesh was less than 5% at identical location of callus region. Therefore, in view of computational efficiency, the medium sized mesh was considered to be sufficiently accurate. The meshing parameters of the rest of the implanted femur was carried forward from author's earlier work [38].

A static load of 2.1kN was applied through the femoral head in such a way that it induces the effect of physiological femur i.e., femur being oriented laterally by 10˚ in the frontal plane and dorsally by 10˚ in the sagittal plane as described for fatigue testing [39]. The physiological load cases replicated maximum load during stance phase of normal walking for a body weight of 80 kg [40,41]. A condition of zero displacement was applied at distal femoral surface nodes sufficiently away from the point of application of load. The screw fastening preload was

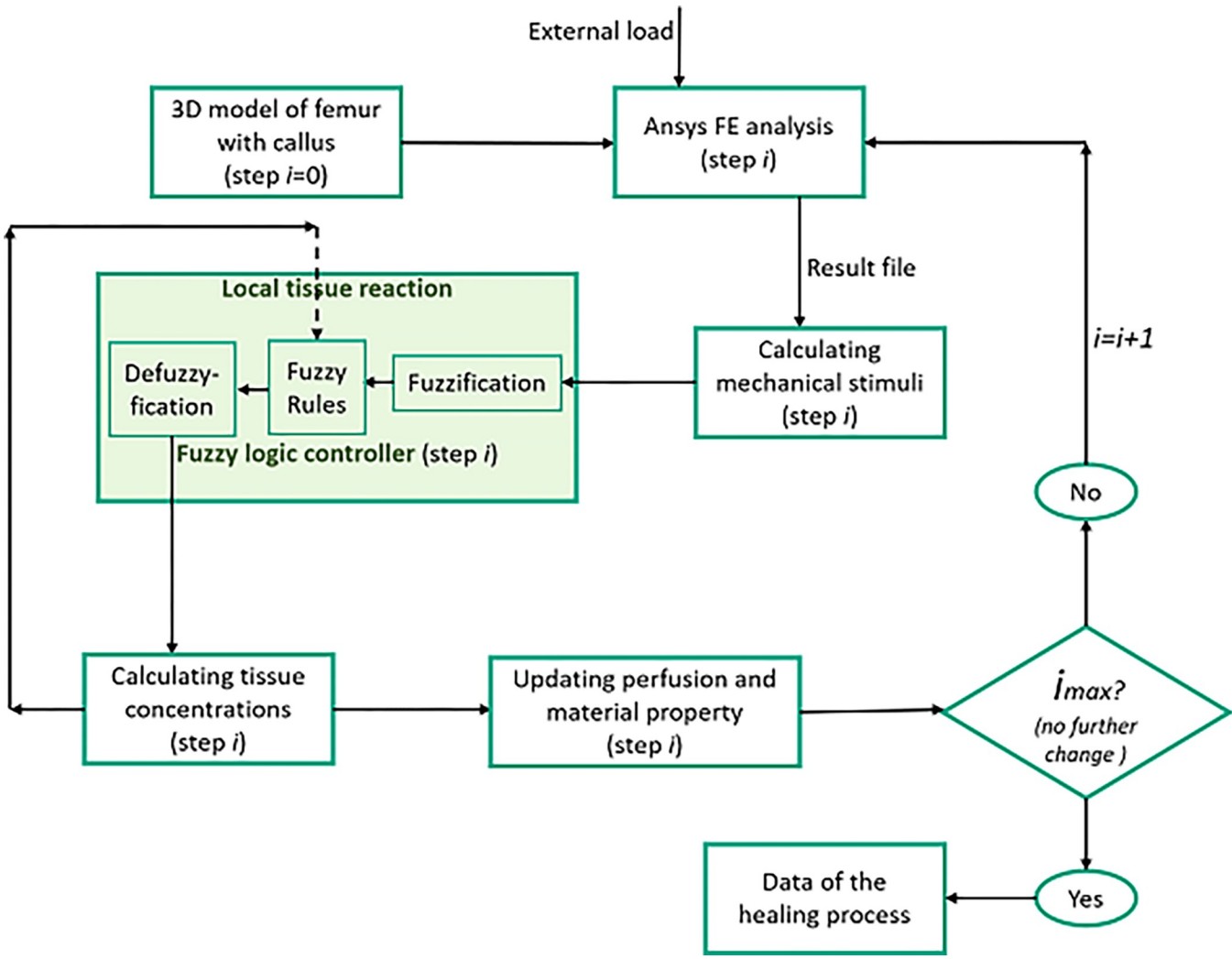

**Fig 1. Flow-chart of dynamic fracture healing simulation model based on fuzzy controller.**

simulated using 'pretension load' by defining the pretension section in APDL for the compression plating. On application of screw-tightening preload, the first thread is found to carry major proportion of load [42]. Therefore, the preload was applied on a slice of the first thread in the screw shaft. The pretension load typically involves normal stresses in the axial direction of the screws. The value of the pretension load was chosen as 500N based on average of some previous studies [6,34,43,44]. As preload is likely to be present for the proximal locking screw for both plating techniques it was not included to highlight the variation between the two plating systems.

The FE models of intact femur was validated based on force-displacement characteristics obtained under in vitro loading (Fig 2D). For experimental test, the intact bone specimen was potted inside a custom-made aluminium rig, after removing the condylar section, as shown in Fig 2D. The custom-made fixture was introduced in order to provide sufficient distal constraint. Identical loading and boundary condition were ensured for validation of the FE model. The regression analysis and student's t-test, revealed a significant correlation between measured and FE predicted curves. A high value of correlation coefficient (R = 0.99) and low

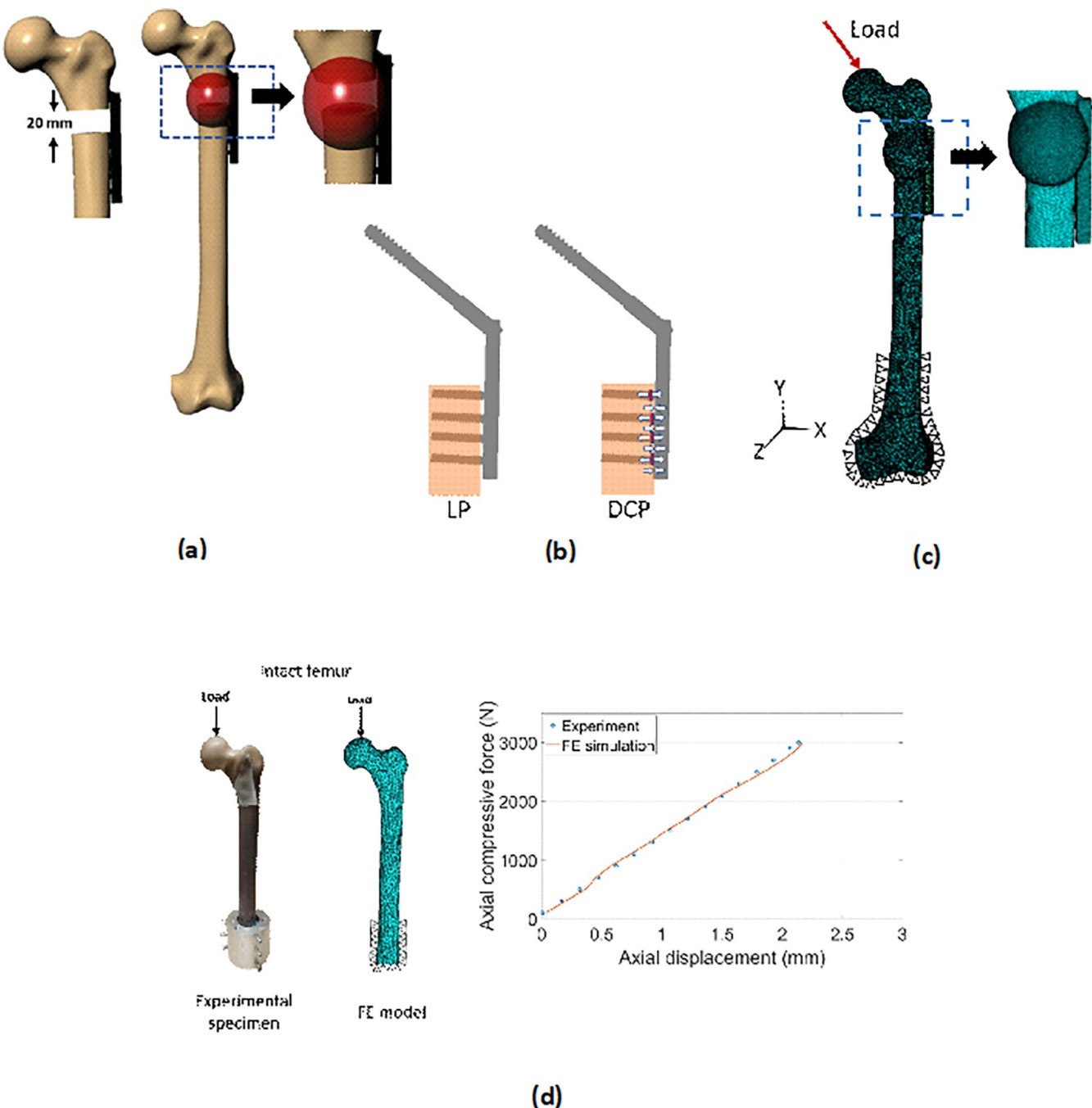

**Fig 2.** Virtual models of the bone constructs and experimental validation: (a) CAD model of femur having fracture gap of 20 mm along with formed callus, (b) CAD models of LP and DCP with depiction of preloads—the red regions indicating preload and white arrows specifying the direction of the preloads and (c) 3D FE model of implanted femur under compressive loading and (d) experimental validation of the intact femur under static compression loading.

standard error of estimate (SE = 0.02 mm) along with a regression slope, b~1.0 (PE of b = 1.4%), and intercept of a = 0.022 were predicted for the intact femur (95% confidence interval). This imparted confidence in the efficacy of the FE model generation scheme which was further employed to develop the plated constructs (LP and DCP).

The material property assigned to bone, callus and implant are presented in Table 1 [18,30,45]. Linear elastic, isotropic and homogenous material properties were considered for cancellous bone, plate and screws. Orthotropic material property was applied to the cortical bone based on the data provided by the manufacturer [45]. The material for the implant was considered to be stainless steel. Poisson's ratio was set as 0.3 for all materials except fibrocartilage (0.45). All interfaces of the FE model for locked type plating were assumed to be bonded under all conditions, whereas for compression plating technique the interface of the distal dynamic screws with plate and bone, and also that between plate and cortical bone was assumed to be in contact with coefficient of friction set as 0.3.

Each callus element comprises its own characteristic material property that were updated after each iteration based on current tissue concentrations. For determination of elements young's modulus (E) and poison's ratio (ν), a mixture rule is applied using the weighted sum of the material properties of the basic tissue types as shown in Eqs 1 and 2 [46].

$$E_{el} = 4000 \text{MPa } C_{el,bone} + 40 \text{MPa } C_{el,cartilage} + 3 \text{MPa } C_{el,connective\ tissue} \quad (1)$$

$$v_{el} = 0.3\ C_{el,bone} + 0.45\ C_{el,cartilage} + 0.3 \text{MPa } C_{el,connective\ tissue} \quad (2)$$

Two independent strain invariants (mechanical stimuli) namely hydrostatic strain ($\varepsilon_{hyd}$) (Eq 3) and distortional strain ($\varepsilon_{dis}$) (Eq 4) were calculated from the principal strains ($\varepsilon_1, \varepsilon_2, \varepsilon_3$) of each element inside the healing region [47]. The hydrostatic strain represents the volumetric change while the distortional strain denotes change in shape.

$$\varepsilon_{hyd} = \frac{1}{3}\left(\varepsilon_1 + \varepsilon_2 + \varepsilon_3\right) \quad (3)$$

$$\varepsilon_{dis} = \frac{1}{\sqrt{2}}\left(\left(\varepsilon_{1-}\varepsilon_2\right)^2 + \left(\varepsilon_{2-}\varepsilon_3\right)^2 + \left(\varepsilon_{3-}\varepsilon_1\right)^2\right)^{1/2} \quad (4)$$

Initially, the callus region was assumed to consist of 100% connective tissue (0% Cartilage and 0% bone).

## 2.2 Biological part: Simulation of tissue differentiation using Fuzzy logic controller

A linguistic fuzzy rule based algorithm was used to simulate the tissue differentiation phenomenon inside the callus region. Fuzzy inference engine was developed using the Fuzzy Toolbox in MATLAB R2019A (The Math Works, Inc., Natick, MA, USA) to predict the tissue differentiation and revascularisation. Mamdani fuzzy inference was introduced to build a control system by incorporating a set of linguistic control rules obtained from experienced human

**Table 1. Material Property data based on ASTM D638[\*\*\*], D695[\*\*], D1621[\*] & 308[#].**

| Material | Young's Modulus (MPa) | Poisson's Ratio |
|---|---|---|
| Cancellous | 155[\*] | 0.3 |
| Cortical | 16,700(*compressive*)[\*\*] 10,000(*transverse tensile*)[\*\*\*] | 0.3 |
| Woven bone | 4000 | 0.3 |
| Fibrocartilage | 200 | 0.45 |
| Connective tissue | 3 | 0.3 |
| Plate & screw (Stainless Steel) | 193,000[#] | 0.3 |

**Input variables**

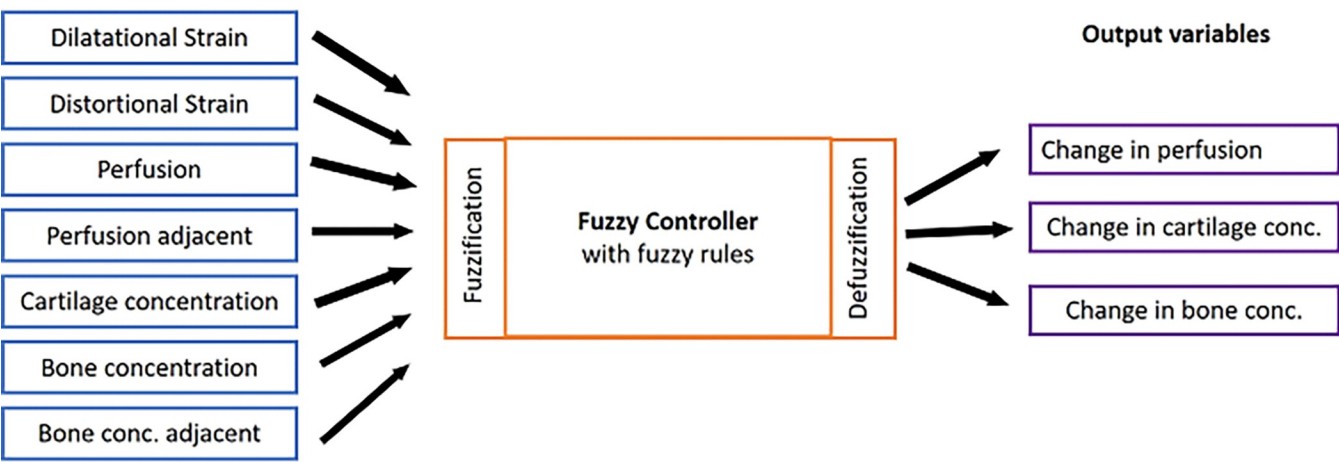

**Fig 3. Fuzzy logic controller of tissue differentiation with seven input and three output variables.**

operators [48]. Mamdani type fuzzy logic controller was employed to link the seven input state variables using some linguistic rules to predict (deterministically) the change in three output state variables (Fig 3). The seven input variables include two mechanical stimuli–hydrostatic strain and distortional strain and five biological variables–perfusion, perfusion in adjacent element, cartilage concentration, bone concentration and bone concentration in adjacent elements. The output variable was change in perfusion, change in cartilage concentration and change in bone concentration.

The fuzzy controller is comprised of 19 linguistic if-then rules that takes into account the process of angiogenesis, intramembranous ossification, chondrogenesis, cartilage calcification, endochondral ossification and tissue destruction (Table 2) [23,24]. Rules no. 1–4 describe the process of angiogenesis, that depends on the local mechanical stimuli and perfusion condition of adjacent elements. Good vascularisation assists direct osteogenesis i.e., intramembranous ossification (Rules 5 and 6) and poor vascularisation distant from the bony tissue favours chondrogenesis (Rules 7 and 8). Rules no. 9–12 describes the process of cartilage calcification which requires high mechanical stimuli irrespective of perfusion. Rules no. 13–16 represents endochondral ossification that happens under medium or high blood perfusion and results in increase in bone concentration and decrease of cartilage concentration. Rules no. 17–19 modelled the overloading condition in bone fracture healing. Destruction of bone and cartilage is observed if the tissue is pathologically overloaded. When active, this rule rapidly decreases the perfusion, cartilage and bone concentration.

The membership function of a fuzzy set is apparently the indicator function in classical sets. In fuzzy logic, it describes the degree of truth as an extension of valuation [49]. Membership function characterizes the information of the fuzzy set and assigns each object a grade of membership ranging between zero and one. Membership function of seven input variables (Fig 4A–4C) and three output variables (Fig 4D–4F) were defined as trapezoidal functions based on theory of Claes and Heigele [8]. Also, the work of Kaspar et al. on cell culture experiments served as the fundamentals for defining the membership functions [50]. These membership functions related the quantative values (strain, concentration, perfusion) to linguistic values (e.g., low, medium, high) and vice versa. The centroid method was used while defuzzification

**Table 2. Tissue differentiation rules describing biological processes within the callus region implemented in the fuzzy controller [29,30].**

| Rule | $\varepsilon_{hyd}$ | $\varepsilon_{dis}$ | $C_{perfusion}$ | $C_{perfusion,}$ neighbour | $C_{bone}$ | $C_{bone,}$ neighbour | $C_{cartilage}$ | $\Delta C_{perfusion}$ | $\Delta C_{bone}$ | $\Delta C_{cartilage}$ | Process |
|---|---|---|---|---|---|---|---|---|---|---|---|
| 1 | Not neg. destructive | About zero | Low | Not low | - | - | - | Increase | - | - | Angiogenesis |
| 2 | Not neg. destructive | About zero | Not low | high | - | - | - | Increase | - | - | Angiogenesis |
| 3 | Not neg. destructive | Low | Low | Not low | - | - | - | Increase | - | - | Angiogenesis |
| 4 | Not neg. destructive | Low | Not low | high | - | - | - | Increase | - | - | Angiogenesis |
| 5 | Neg. low | Low | High | - | - | High | Low | - | Increase | - | Intramembranous ossification |
| 6 | Pos. low | Low | High | - | - | High | Low | - | Increase | - | Intramembranous ossification |
| 7 | Neg. medium | Not. destructive | - | - | - | - | - | - | - | Increase | Chondrogenesis |
| 8 | Neg. low | Not destructive | - | - | - | - | - | - | - | Increase | Chondrogenesis |
| 9 | Neg. medium | Not destructive | - | - | - | Not low | Not low | - | Increase | Decrease | Cartilage calcification |
| 10 | Neg. low | Not destructive | - | - | - | Not low | Not low | - | Increase | Decrease | Cartilage calcification |
| 11 | About zero | Not destructive | - | - | - | Not low | Not low | - | Increase | Decrease | Cartilage calcification |
| 12 | Pos. low | Not destructive | - | - | - | Not low | Not low | - | Increase | Decrease | Cartilage calcification |
| 13 | Neg. low | Zero | Not low | - | High | High | Low | - | Increase | Decrease | Endochondral ossification |
| 14 | Neg. low | Low | Not low | - | High | High | Low | - | Increase | Decrease | Endochondral ossification |
| 15 | Pos. low | Zero | Not low | - | High | High | Low | - | Increase | Decrease | Endochondral ossification |
| 16 | Pos. low | Low | Not low | - | High | High | Low | - | Increase | Decrease | Endochondral ossification |
| 17 | Neg. destructive | - | - | - | - | - | - | Decrease | Decrease | Decrease | Tissue destruction |
| 18 | Pos. destructive | - | - | - | - | - | - | Decrease | Decrease | Decrease | Tissue destruction |
| 19 | - | Destructive | - | - | - | - | - | Decrease | Decrease | Decrease | Tissue destruction |

i.e., the final output prediction was an average sum of the weighted single output of the active rules.

The initial perfusion condition was defined in accordance with the work of Simon et al. [30]. Cortex element was assumed to have intact vascularity (100% blood perfusion) excluding the areas adjacent fracture gap. The remaining end of these fragments were defined as initially avascular i.e., 0% blood perfusion. Perfusion was allowed from cortical bone fragment as well as from the peripheral and medullary boundaries to the callus region [51]. At the peripheral boundary of the callus, the perfusion was set to 30% which represents the 'extraosseous blood supply' from adjacent soft tissues [51]. After 10 days, the perfusion B.C. in the medullary channel was set to 30% to represent potential revascularisation from the marrow.

## 3. Results

The computational work demonstrated the stages of callus healing in two types of plating techniques, namely locking and dynamic compression plate. Our simulations predicted the bone concentration, cartilage concentration, blood perfusion and change of modulus in the callus region over space and time, for both cases of LP and DCP, using FE analysis and fuzzy logic-based preclinical models. We further predicted tissue concentrations and distortional strain values based on the outcomes of iterative simulation steps by considering each iteration as one day. The simulation was continued iteratively till 60 healing steps running in loop. We

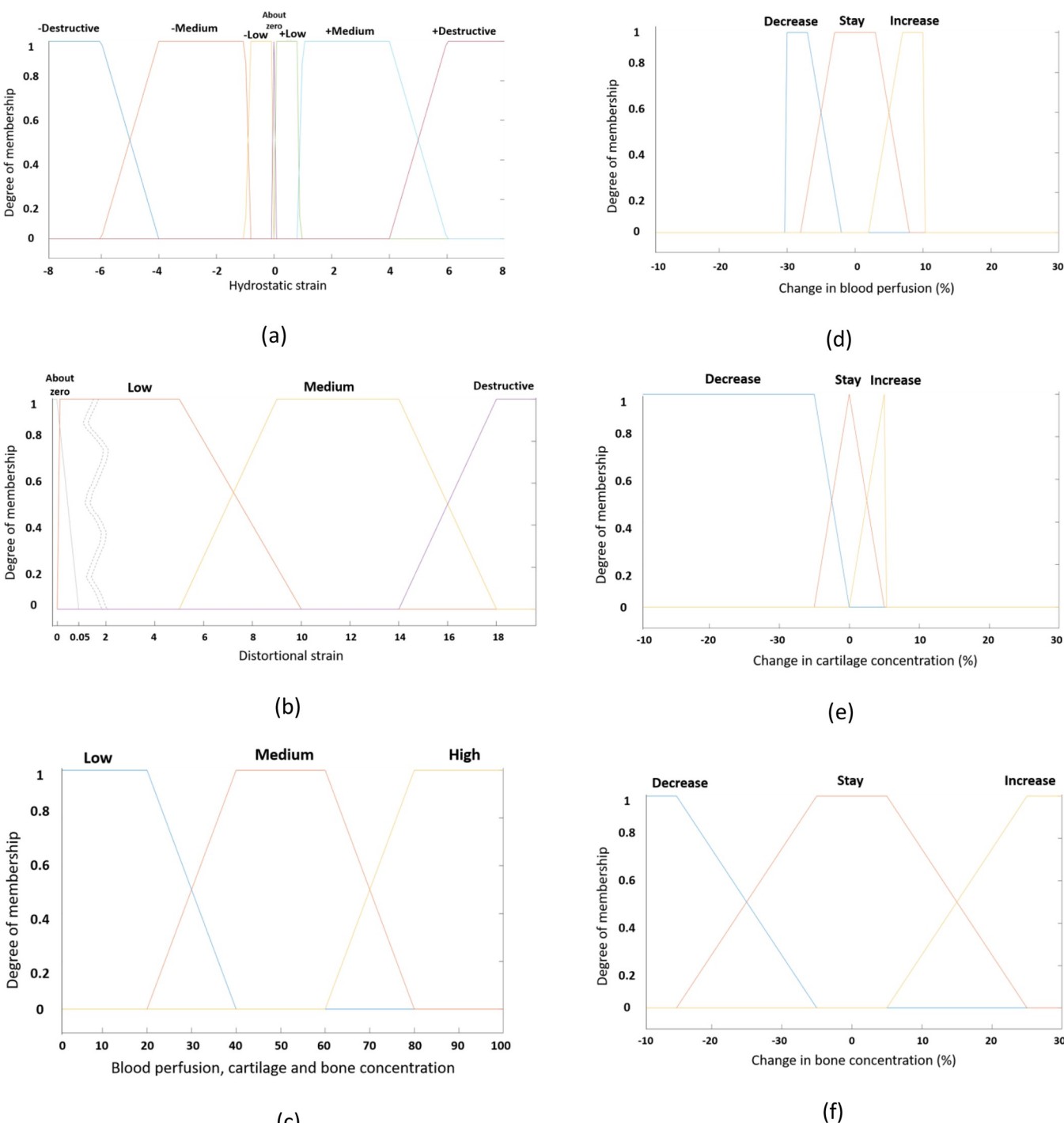

**Fig 4.** Membership functions: (a) and (b) were used to describe the first two input variables, e.g. hydrostatic strain and distortional strain, respectively, whereas (c) was used for the remaining five input variables, e.g. blood perfusion, perfusion in neighbour elements, cartilage concentration, bone concentration and bone concentration in neighbour elements. Membership functions (d), (e) and (f) were used to describe the three output variables, e.g. change in perfusion, change in cartilage concentration and change in bone concentration, respectively.

calculated the connective tissue concentration by subtracting the summation of bone and cartilage from the total tissue concentration (i.e. 100%). The magnitude of strain experienced by the fracture region primarily depends on the applied load and the stiffness of the device, and in

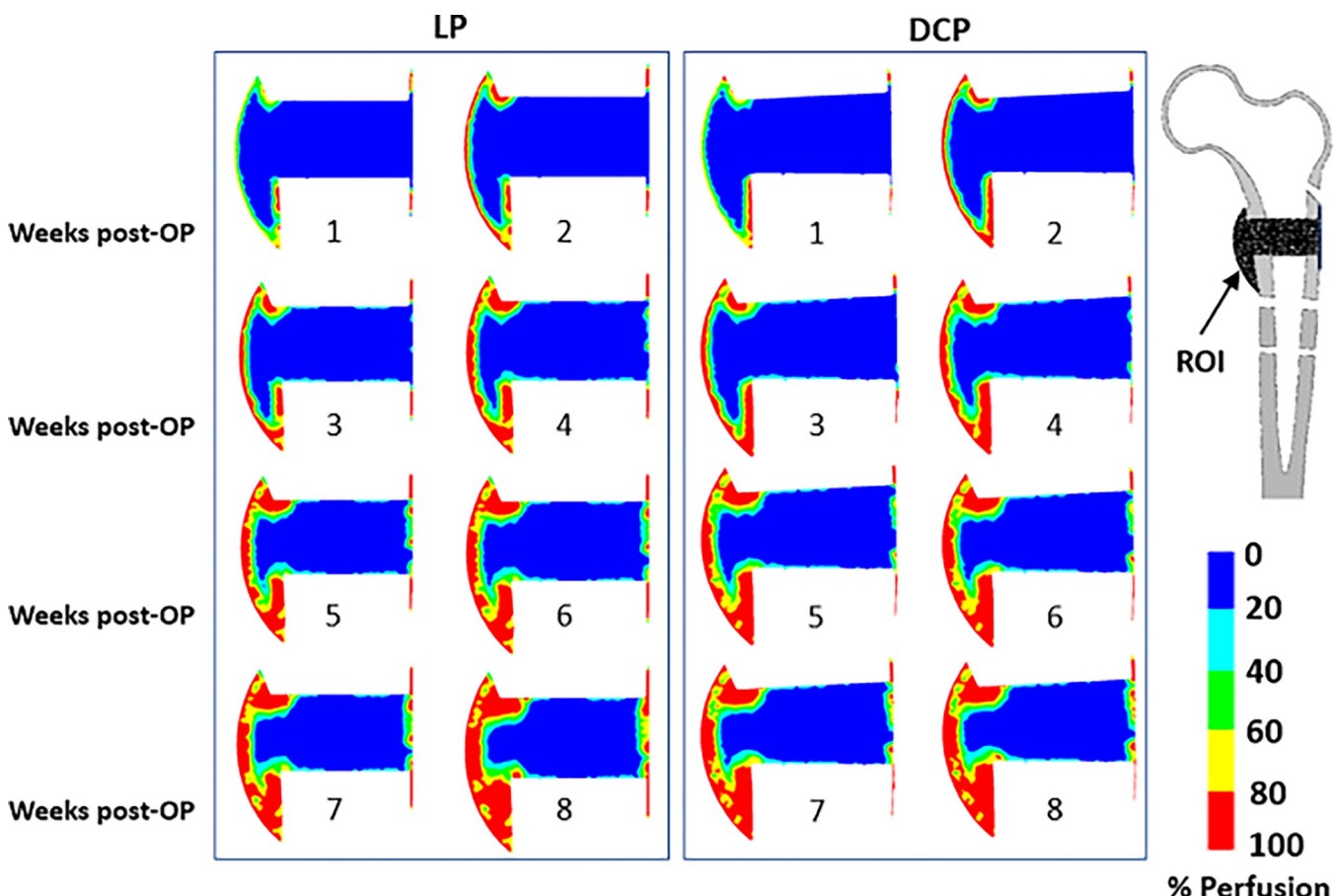

**Fig 5. Post-operative distribution of percentage blood perfusion in the ROI from week 1–8 for (a) LP and (b) DCP.**

turn determines the tissue differentiation pattern of the region. The qualitative predictions of tissue differentiation patterns and other healing parameters in the callus region are showcased through contours in the longitudinal cross-section, considered hereinafter as the region of interest (ROI).

Blood perfusion started at the cortex away from the fracture gap at peripheral side for both cases of LP and DCP, as shown in the ROI (Fig 5). Gradually the blood perfused towards the gap centre. At the beginning of the 4th week, blood perfusion at the periosteum callus was completed for both the cases. Conversely, no revascularisation at the mid-callus was predicted till the end of 8th week.

In the early phases, we observed cartilage formation in the areas with high mechanical stimuli and low blood perfusion for both LP and DCP (Fig 6). Around the 5th week, we estimated the cartilage concentration to be the highest in both cases (~73%). High mechanical stimuli in the middle of fracture gap activated endochondral ossification where cartilage formation took place. This later transformed into woven bone resulting in delayed healing.

At the early stages of healing, we could observe bone formation at the surfaces of cortical bone away from the fracture gap (Fig 7). Gradually from the 6th week onwards, bone formation propagated into the peripheral healing region where bony bridging occurred. Finally, the entire callus got transformed into newly formed woven bones (8th week). For both LP and DCP, the initial commencement of bone formation was through intramembranous

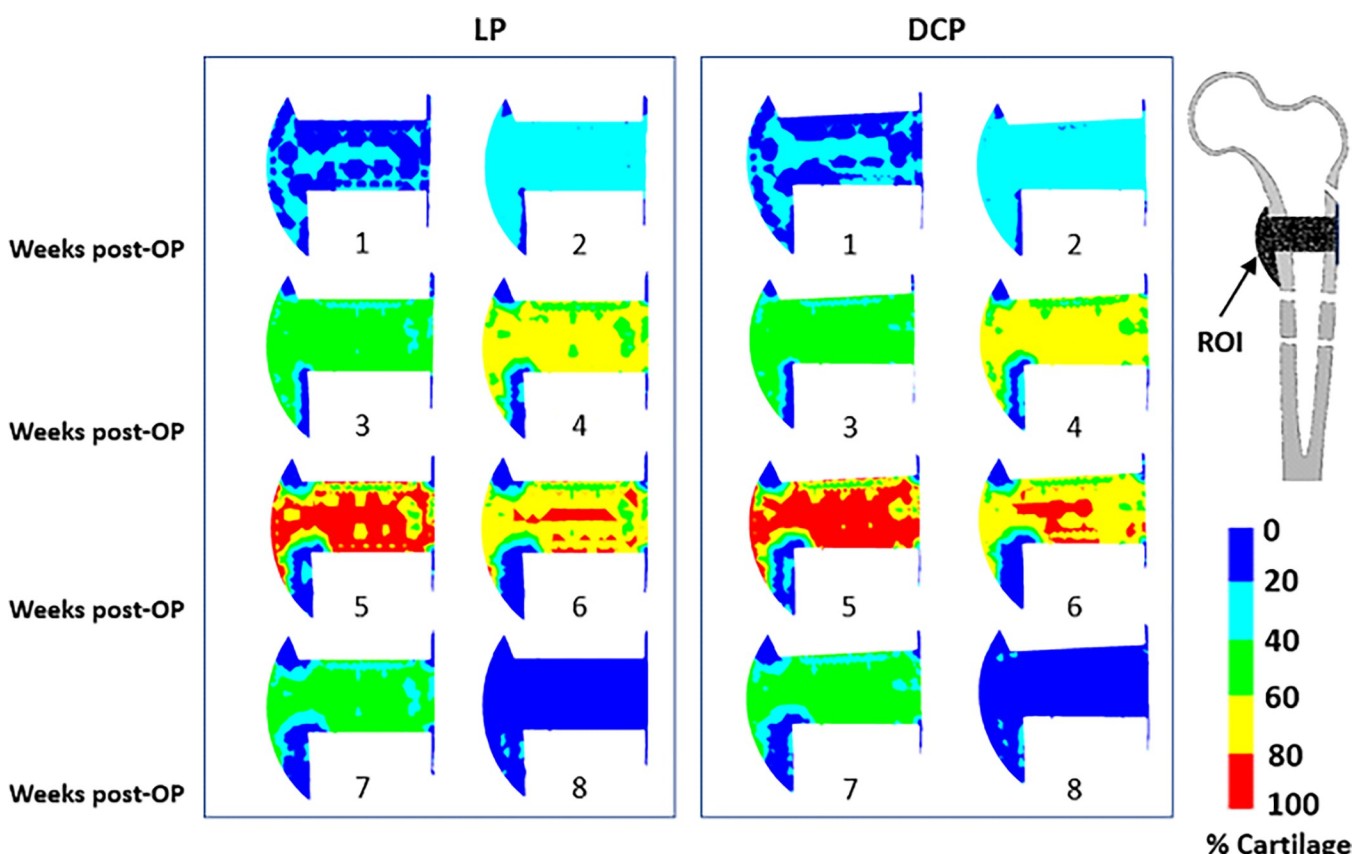

**Fig 6. Post-operative distribution of percentage cartilage concentration in the ROI from week 1–8 for (a) LP and (b) DCP.**

ossification. However, as the healing progressed, endochondral ossification was dominant between fractured bone and external to periosteal callus. In contrast, intramembranous ossification occurred in the callus directly in contact to the distal and periosteal end of the cortex.

The overall change in modulus of the callus region is presented in Fig 8. For both cases of LP and DCP, although the modulus increased around same time, we observed no significant variation in magnitude. However, callus modulus increased more for LP as compared to DCP at the early phases of healing (10.38%). Nonetheless, in the later stages, i.e. 4th week onwards, DCP was associated with a greater callus modulus. The simulations predicted an overall similar trend in bone healing as well (Fig 8).

By the end of 5th week, there were almost no trace of connective tissues for both the cases (Fig 9A and 9B). Also, woven bone concentration for both cases at the end of 4th week was around 13% which reached almost 99% after 8th week (Fig 9A and 9B). Our simulations further predicted higher distortional strain at the initial stages in the case of DCP as compared to LP (Fig 10A). From 3rd week onwards, the callus strains gradually decreased and at later stages, we observed no significant difference thereof between LP and DCP.

## 4. Discussions

The present study employed a biomechanical model to predict tissue transformation during different stages of secondary fracture healing associated with subtrochanteric femur fracture. The purpose of this study was to understand the difference in fracture healing scenarios at the callus region for two implantation techniques, namely LP and DCP. Fuzzy rules acquired from

medical observations were implemented to describe biological phenomena of angiogenesis, intramembranous or endochondral ossification and cartilage calcifications. The entire simulation was carried out in batch mode using a workstation (Model: HP Z2 TWR G4 / OS: Windows 10 Pro / Proc: Intel® Core™ i7-9700 3.0 GHz 8-core CPU / RAM: 32 GB), where each iteration took roughly 1 hour to complete.

The overall blood perfusion patterns (Fig 5) were predicted to be similar for both cases of LP and DCP, and are consistent with the earlier histological observations of Rhinelander et al. [43]. A large area of the mid-callus inside the endosteum remained avascular for a long computational time due to the experience of high distortional strain. Within the simulated time frame of the 8[th] week, the blood perfusion did not fill the interfragmentary gap completely, which agrees with some of the previous observations [51,52]. The delay in revascularisation in the gap region resulted in slowing down of the ossification process. This region was found to be dominated by formation of cartilages instead of woven bone owing to high mechanical stimuli and low blood perfusion (Fig 6). Claes et al. also reported similar observations in the interfragmentary regions [53]. In those areas, connective tissue persisted as late as in the 4[th] week primarily due to high mechanical stimuli (Figs 7 and 8). The new woven bone formation occurred about two weeks later at the periosteal surface away from the fracture gap (Fig 7). This perhaps may be ascribed to the direct formation of woven bone through the biological process of intramembranous ossification, which requires adequate blood perfusion and the presence of existing bony surfaces.

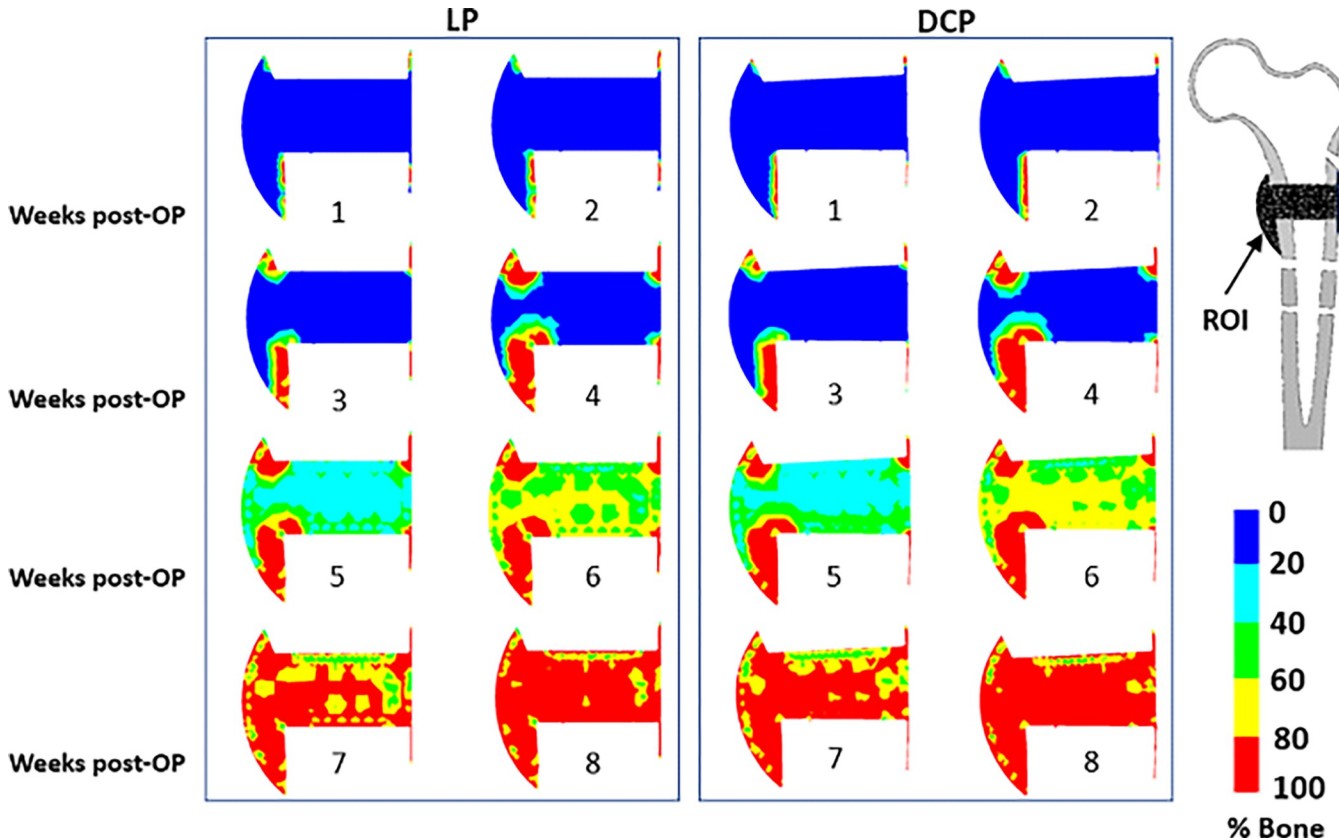

**Fig 7. Post-operative distribution of percentage bone concentration in the ROI from week 1–8 for (a) LP and (b) DCP.**

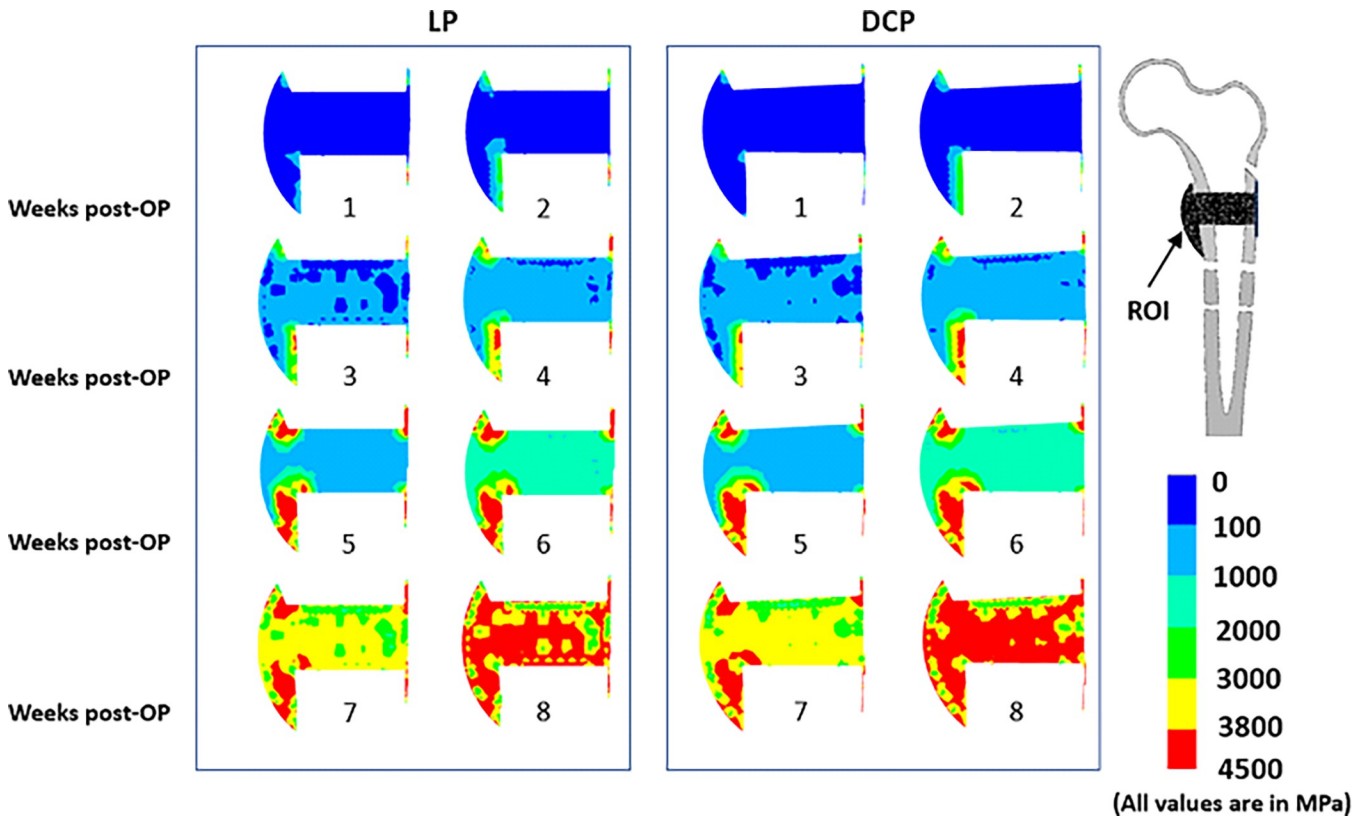

**Fig 8.** Post-operative distribution of modulus (unit MPa) in the ROI from week 1–8 for (a) LP and (b) DCP. It may be noted that the modulus gives a measure of mechanical strength of the region (i.e. higher the modulus, stronger is the region).

As the healing process continued, the dominant type of ossification shifted to endochondral ossification from intramembranous ossification. Once the peripheral bone formation was completed, the strains in the middle of the interfragmentary gap decreased significantly, and bridging started almost at the 6[th] week. The bridging occurred mainly through endochondral ossification. Various studies also reported occurrences of endochondral ossification in the far cortex zone of callus, while intramembranous ossification occurred in the near cortex zone [54,55]. This accorded well with our findings. Further, the degree of calcification predicted in the present study in the near cortex is different from that of the far cortex (Fig 8). This is in accordance with the clinical findings of Lujan et al., wherein they reported asymmetrical callus formation in distal femur fracture implanted with LP [56].

The present study further predicted that compressive strength imparted by the DCP helps in the endochondral ossification at the later stages of healing and facilitates faster solidification of callus as compared to the LP (Figs 7 and 8). In DCP technique, the physiological load experienced by the femur is transferred as frictional load at the plate-bone interface, whereas in LP, it is transferred through the screw-bone interfaces. Thus, the local environment in the bone fracture healing region around screws is influenced by a mix of effects, unlike the straightforward mechanism related to LP. Also, DCP requires a relatively extensive surgical approach and contributes to necrosis, consequently enhancing the risk of delayed callus formation [57].

In the present study, the rapid development of callus was observed between week 4 and week 8 postoperatively (Fig 9A and 9b) for both the cases of LP and DCP, which agreed well with the findings of Gardener et al. [58]. The initial distortional percentage strain associated

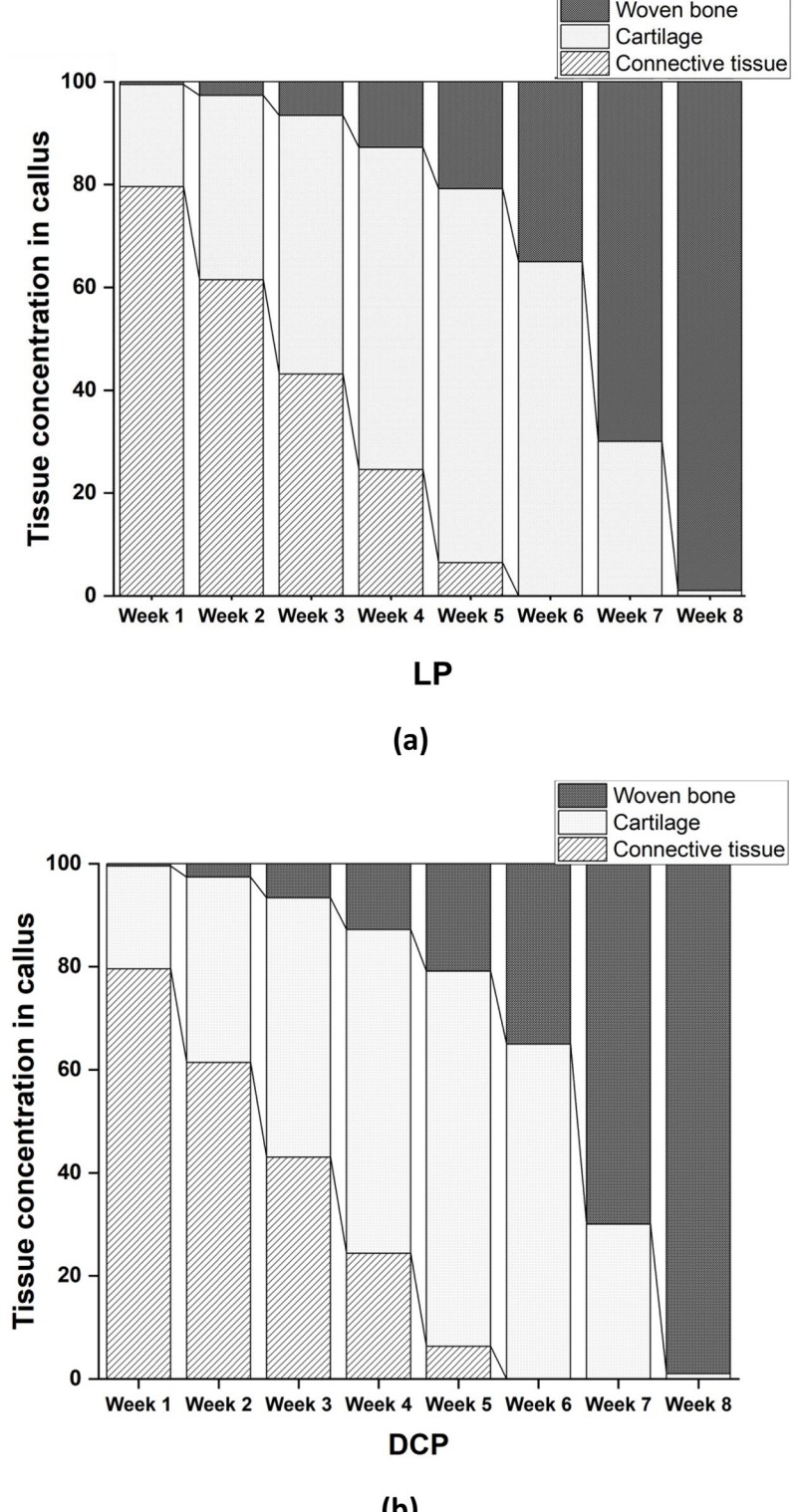

**Fig 9. Percentage variation of various tissue concentrations in the callus region from week 1–8 for (a) LP and (b) DCP.**

with LP and DCP in the callus region was found to be 3.2% and 3.6%, respectively (Fig 10A), which is ideal for callus growth. Strain ranging from 2% to 10% was also reported to facilitate effective generation and development of callus in the fracture site [3]. The gradual decrease of the distortional strain for both the cases in the present study (Fig 10A) matches the trend of the gradual decrease of interfragmentary movement reported in the *in vivo* experiments of Claes et al. [59]. The difference of initial distortional strain in the callus region of LP and DCP fixated femur was primarily responsible for the characteristic variation in the healing response. Hence, an initial delay in healing associated with DCP was predicted when compared with LP (Fig 8). However, at later stages after the initial calcification were achieved, healing was predominant for DCP fixated femur owing to the compressive stress environment in the callus region. Although identical progression of healing for both the cases was predicted after the 8[th] week, early stabilisation of the callus for LP was observed. Ghimire et al. reported that the increase of flexibility in the LP system potentially promotes uniform bone formation across the fracture gap, resulting in better healing outcomes [60]. Various other studies also advocated using flexible fixation to induce the fast formation of callus [61,62]. It may be noted that the LP technique tolerates some degree of mobility in the interface of the fracture essential for quick healing. The interfragmentary movements (IFM) at different healing intervals were compared with that reported by two different studies–one numerical mechanobiology based study Lacroix et al., 2002 [63] and the other an animal experiment carried out by Claes et al. [64] (Fig 10B). Whereas the numerical model [63] demonstrated identical trend in bone healing progression as predicted by our scheme, a significant drop in IFM was measured in the animal experiment [64] as the bone healing progressed beyond 4th week post-operatively. It may, however, be noted that the starting point of the IFM characteristics varied significantly owing perhaps to the differences in model and mechanical input parameters (stimulation). This further illustrates the dependence of the bone healing progression on the input parameters, e.g. mechanical strain.

The above similarities notwithstanding, the study had certain limitations. The musculoskeletal load applied in the present study was assumed to be static and full weight-bearing over the entire healing time in contrast to the dynamic and partial weight-bearing nature as seen in reality. However, this assumption can be justified considering the predicted distortional strain was in the safe range, indicating that the maximum movement was reached in early weeks in both cases. The future scope of study may include the cyclic nature of loading. Bone resorption and cortical remodelling had not been considered to accurately predict callus shape and size in the later phase of healing. These processes were ignored since the primary focus was on the healing process during the early phases till callus bridging. Moreover, the poroelastic material properties were not considered in the current study. The reason for this was the application of axial load in the model. Consequently, considering hydrostatic strain as mechanical stimulus was more appropriate as compared to flow parameters. The material properties assigned to biological soft tissue were assumed to be linear instead of viscoelastic and nonlinear. However, not enough experimental data are available for such implementations.

FE analysis is an approximate method which is sensitive to parameters like loading, material properties and boundary conditions. Output results may vary considerably, thereby necessitating experimental validation of the FE scheme, like the one the present study has conducted. Also, there exists no precise relationship between the iterative time steps involved, and actual time in days or weeks. Therefore, animal study is warranted at the least, with precise estimation of the loading regime such that a full-fledged validation of the present scheme can be contemplated.

Nevertheless, the present preclinical model is robust enough to predict bone healing for different implantation techniques, and thus can serve as a non-invasive tool for evaluating the

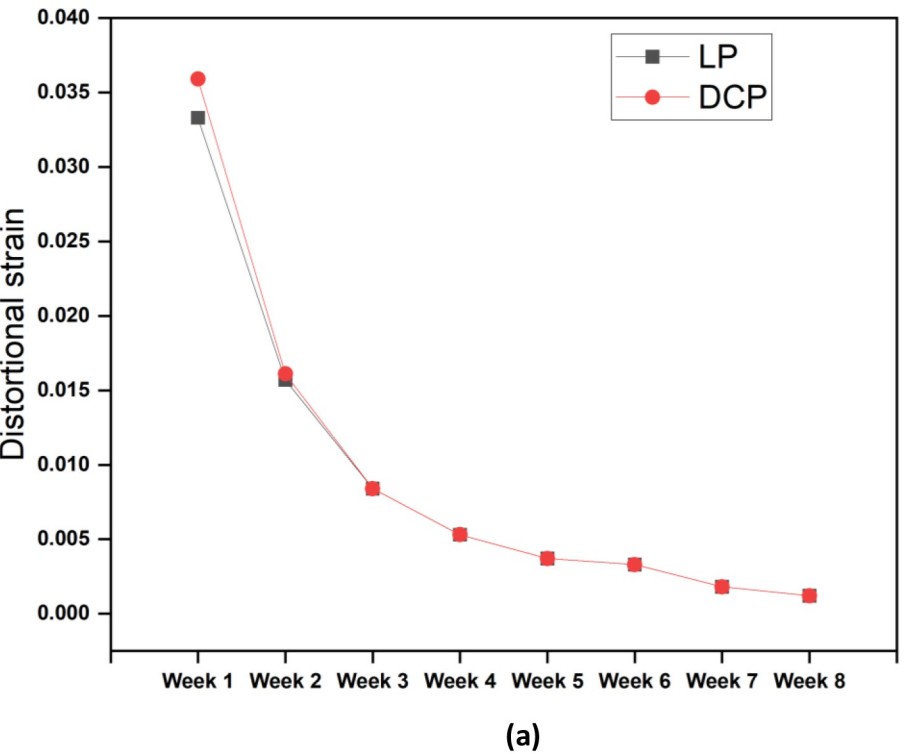

**(a)**

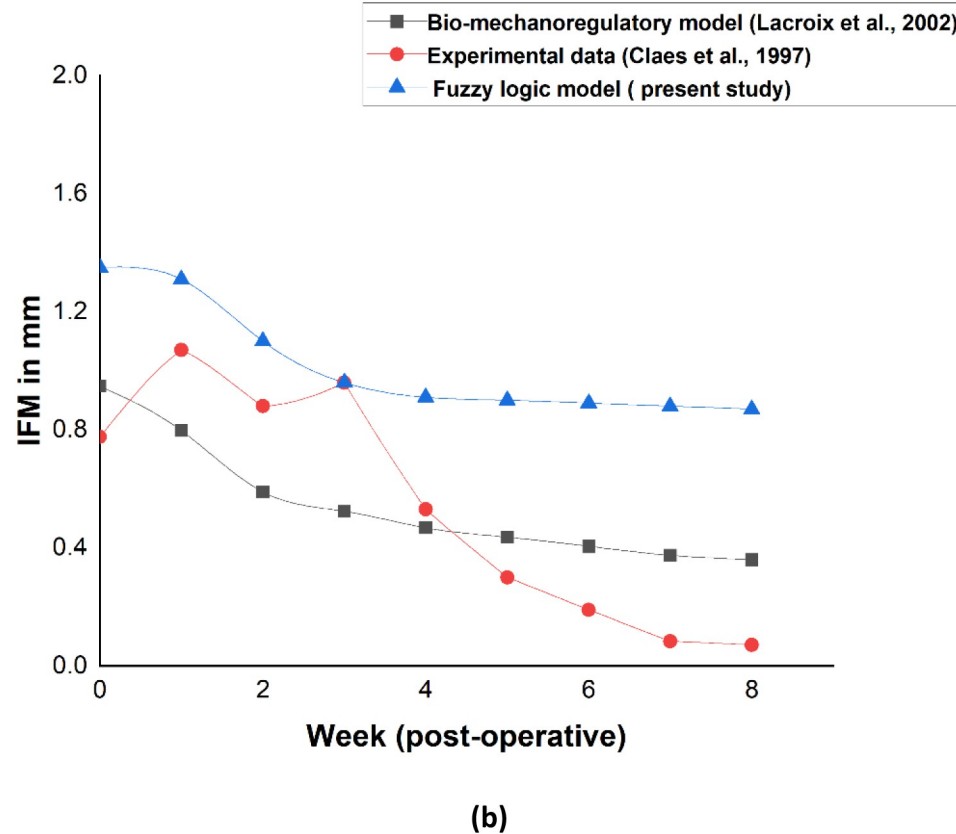

**(b)**

**Fig 10.** Distortional strain and interfragmentary movement (IFM) assessment: (a) distortional strain in the callus region corresponding to LP and DCP constructs as predicted by the present study and (b) the predicted IFM values in LP compared with existing literature.

relative merits of these implants. Rule-based models, e.g. fuzzy logic tools are relatively easier to implement and require less computational time as compared to the equation-based mechanoregulatory models. Furthermore, such techniques provide additional flexibility of incorporating any new rule without disruption of the original model. After incorporating various biological and patient-specific parameters, the fuzzy logic-based iterative model can further account for different clinical phenomena, such as smoking, diabetes, etc. Additionally, the linguistic rules can easily be altered as per the understanding of the medical experts for a more rigorous preclinical assessment. While plan of action for traditional treatment of fractures depends primarily on the experience of the orthopaedist, the present model can act as a platform to predict post-surgery effects in advance and, thereby help clinicians strategize the best implantation technique for a patient. The model can also be explored/applied in veterinary fractures which are, physiologically and in various aspects, similar to those occurring in human patients.

## Supporting information

**S1 File.**
(DOCX)

## Acknowledgments

The authors would like to acknowledge the computational facilities available at the Biomechanics Laboratory, Biosciences and Bioengineering, Indian Institute of Technology Guwahati, India, which has helped to carry out this research study. The authors would also like to acknowledge Dr. Bhaskar Borgohain, Orthopaedic Surgeon, North Eastern Indira Gandhi Regional Institute of Health & Medical Sciences, Shillong, India, for his invaluable inputs.

## Author Contributions

**Conceptualization:** Souptick Chanda.

**Funding acquisition:** Souptick Chanda.

**Investigation:** Pratik Nag.

**Methodology:** Pratik Nag.

**Project administration:** Souptick Chanda.

**Resources:** Souptick Chanda.

**Software:** Pratik Nag.

**Supervision:** Souptick Chanda.

**Visualization:** Pratik Nag.

**Writing – original draft:** Pratik Nag.

**Writing – review & editing:** Souptick Chanda.

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
