## [Decision Letter · Decision Letter 0]

18 Feb 2022

PONE-D-21-37870Secondary bone healing for subtrochanteric femoral fracture treated with locking and dynamic compression plates: a comparative phenomenological assessment based on fuzzy interpretationsPLOS ONE

Dear Dr. Chanda,

Thank you for submitting your manuscript to PLOS ONE. After careful consideration, we feel that it has merit but does not fully meet PLOS ONE’s publication criteria as it currently stands. Therefore, we invite you to submit a revised version of the manuscript that addresses the points raised during the review process.

We look forward to receiving your revised manuscript.

Kind regards,

Yiming Tang, Ph.D.

Academic Editor

PLOS ONE

Reviewers' comments:

Reviewer's Responses to Questions

**Comments to the Author**

1. Is the manuscript technically sound, and do the data support the conclusions?

Reviewer #1: Partly

Reviewer #2: Yes

2. Has the statistical analysis been performed appropriately and rigorously? 

Reviewer #1: No

Reviewer #2: Yes

3. Have the authors made all data underlying the findings in their manuscript fully available?

Reviewer #1: Yes

Reviewer #2: Yes

4. Is the manuscript presented in an intelligible fashion and written in standard English?

Reviewer #1: No

Reviewer #2: Yes

5. Review Comments to the Author

Reviewer #1: General Comments

In the manuscript by Nag and coworkers, the authors have developed a theoretical assessment of secondary bone healing when treated with either locking plates or dynamic compression plates. The authors use fuzzy logic-based interpretations to comparatively asses the two methods of treatment. But the methods and results are not well described. Considering the broad readership of Plos One, the authors should describe the methods and their results clearly. Overall, the language used in the manuscript is very ambiguous and makes the manuscript difficult to read. The authors have predicted the healing of the bone by simulating the mechanical compression of the bone using finite element modeling and tissue differentiation using fuzzy logic methods. But these predictions are not supported by any experiments. Without any experimental data, the claims of the authors are not well supported. I do not recommend this manuscript for publication in the current form. But if the authors make significant changes in the text and add some experimental data, the manuscript could add to the current understanding of bone healing.

Major comments:

1. The title of the manuscript is very long and confusing. The authors should make their title concise and clear. They can use the short title as their main title.

2. In the second paragraph of introduction, the author state that “The biomechanical environment achieved using LP is totally different form that obtained through DCP”. But they do not provide any references for this statement and they also do not elaborate on this statement.

3. In the introduction the authors do not highlight a clear gap in the current understanding, making it difficult to assess the importance of their work. The authors should clearly state the hypothesis at the end of the introduction

4. In the biological part of the methods section, the authors write about fuzzy logic based methods. But they do not explain “Mamdami type fuzzy logic controller” and do not provide any reference. Further in the same section, they bring in the concept of Membership function. But these functions are not clearly defined and explained. Without properly explaining these functions, it is rather difficult to comprehend the results

5. The text in the results section is very ambiguous. It is not clear whether the authors are describing an experimental data or a prediction of their simulation.

6. The figures are not prepared carefully and the figure legends do not describe the figure well.

Minor comments

1. The manuscript is hard to read as the authors use passive voice throughout the manuscript. The authors should try to write more in active voice.

2. In figure 3b, it is not clear what is the meaning of destructive and medium.

3. It would be better to make the plots in figure 3 smaller, so that all the plots fit in one page. This makes it much easier to understand.

4. In Fig. 4, it is not mentioned what the color bar represents.

5. Similarly, in Fig. 5 and Fig. 6 description of color bar is missing.

6. In Fig. 8, it is not mentioned whether tissue concentration and distortional strain is a mean of multiple simulations or a single simulation. It is also not clear how many simulations were performed. Moreover, it would be better if the authors bring all the plots of Fig. 8 in one page.

7. The authors should put more effort in organizing the figures and annotating them.

Reviewer #2: A well-written manuscript wherein the study predicts and compares the healing process of a subtrochanteric femoral fracture step by step in response to two different fixation techniques, the Locking Plate (LP) and the Dynamic Compression Plate (DCP), using the fuzzy logic-based iterative model. Authors here hypothesized that the different screw fixation mechanisms seen in these two techniques might influence the healing outcome differently. Final results suggested an initial delay in response with DCP but an altogether identical progression of healing for both the cases after the 8th week time-point along with an early stabilisation of the callus for LP. All these findings were in accordance with previous studies done using different tools.

Thus, this study brings to the limelight that fuzzy logic controls could actually be used to study and predict tissue transformation in a realistic and non-invasive manner. In fracture management, it can help to foresee the probable course of healing process and compare the expected outcomes with the use of different types of fixation plates prior to the surgery itself. This will help to select the best option available for the patient based on his individual characteristics and fracture types beforehand.

• Reference missing in the first paragraph of the introduction.

• Various authors have observed that LP performs better than DCP in older or weaker bone [1,5]. Uhl et al. found that DCP provides inherently more stability than LP [6]. Gap closure and bone-to-bone load transfer resulted in better mechanical performance of DCP over LP during axial compression [6]. The author claims various authors and cited only one. Add few more citations to claim LP better than DCP.

• What was the values obtained for mesh sensitivity analysis?

• Will there be any changes in the parameter if the loading was done cyclic rather than static loading?

• Add a paragraph on the limitation of the study and FEA.

• Could you please add a note on why the tools used in this study are better than other comparison tools used elsewhere?

• Please elaborate on the possible in-future benefits of this study?

• Could you please add a brief concluding paragraph towards the end?

6. PLOS authors have the option to publish the peer review history of their article (what does this mean?). If published, this will include your full peer review and any attached files.

Reviewer #1: No

Reviewer #2: **Yes: **Hemant V. Unadkat

---

## [Author Response · Author response to Decision Letter 0]

17 Mar 2022

A point-to-point response file is attached addressing all queries/comments of the editor and reviewers.

---

## [Decision Letter · Decision Letter 1]

9 May 2022

PONE-D-21-37870R1Phenomenological prediction of post-surgery secondary bone healing for subtrochanteric femoral fracture based on fuzzy interpretationsPLOS ONE

Dear Dr. Chanda,

Thank you for submitting your manuscript to PLOS ONE. After careful consideration, we feel that it has merit but does not fully meet PLOS ONE’s publication criteria as it currently stands. Therefore, we invite you to submit a revised version of the manuscript that addresses the points raised during the review process.

We look forward to receiving your revised manuscript.

Kind regards,

Yiming Tang, Ph.D.

Academic Editor

PLOS ONE

Journal Requirements:

Reviewers' comments:

Reviewer's Responses to Questions

**Comments to the Author**

1. If the authors have adequately addressed your comments raised in a previous round of review and you feel that this manuscript is now acceptable for publication, you may indicate that here to bypass the “Comments to the Author” section, enter your conflict of interest statement in the “Confidential to Editor” section, and submit your "Accept" recommendation.

Reviewer #1: All comments have been addressed

Reviewer #2: All comments have been addressed

Reviewer #3: All comments have been addressed

2. Is the manuscript technically sound, and do the data support the conclusions?

Reviewer #1: Partly

Reviewer #2: Yes

Reviewer #3: Yes

3. Has the statistical analysis been performed appropriately and rigorously? 

Reviewer #1: No

Reviewer #2: Yes

Reviewer #3: N/A

4. Have the authors made all data underlying the findings in their manuscript fully available?

Reviewer #1: Yes

Reviewer #2: Yes

Reviewer #3: Yes

5. Is the manuscript presented in an intelligible fashion and written in standard English?

Reviewer #1: No

Reviewer #2: Yes

Reviewer #3: Yes

6. Review Comments to the Author

Reviewer #1: The authors have answered most of the comments. Though authors did not perform experiments, they compared their simulation data with existing experiments. As mentioned by the authors, performing experiments could be a complete work on its own. But the language used in the manuscript is still hard to comprehend. I would recommend publication only if the authors address the minor comments.

Minor comments:

1. The figures have improved from the submitted version. But the figure legends are too short and do not describe the figures well. This particularly true for Figs. 5-9.

2. Authors mostly use passive voice to describe the results. In general Passive voice is used to describe the methods. But it would be best to use active voice to describe the results. The article below uses active voice for describing results.

https://www.nature.com/articles/s41567-018-0358-7

I hope that the authors would agree that this adds to clarity of the manuscript.

Reviewer #2: This reviewer has strong objection to the word "Phenomenological" in the title. Can this word be omitted from the title? The dictionary meaning of the word is "relating to the science of phenomena as distinct from that of the nature of being." as such it being distinct from the nature of being is misleading in the title.

Reviewer #3: To set the context of my review, I first read the original article (without previous comments or suggestions from other reviewers). I then added my own observations/impressions and finally, read the concerns raised by both reviewers, which I found sensible and needed. I finally went (after having finished my revision) onto read the amended manuscript, which reads much better than the original.

I found this manuscript interesting, and easy to follow (with a few exceptions on the language and jargon). Possibly, due to my background in both clinical and research settings with both human and animal patients, that helped me to perceive the value of this study, as long as we acknowledge that this is a ‘model’ (and modelling is very powerful and valuable in many aspect, but in some instances, clinicians they either lack the understanding of scientific literature of find hard to give credit to models since they work with ‘real’ cases).

In this work, the authors set the context of fractures (e.g. causes, importance, etc) which affects different populations. The importance of various elements for fracture healing (stability, bone healing, haemorrhage, callus formation, stages, etc) is also broke down in the context of the primary (1) and secondary (2) bone types of healing.

The key role of the position of the screws in both types of plates (DCP being the standard) is discussed, the authors introduced the new type of plate proposed and analysed in their study, i.e. locking plate (LP), and the benefits of this in regards to screw position and fixation and in comparison to DCP, arguably better than LP, according to some previous works. Also, other predictive models and their pros and cons are accurately presented and discussed in detail. To complete, they highlighted the lack of studies comparing overall progression of fracture healing derived from the effect of ‘screw type’. Outstanding is the fact that most models presented are, in ‘primitive 3D models’ (the authors’ opinion) instead of ‘real’ bone and implants.

The authors’ interest is to demonstrate the predictive power of applying non-invasive fuzzy logic controls to determine tissue transformation and (secondary) healing in plate bones. With a particular interest, as they noted, in the influence of the type of screw fixation. I here would stress that the manuscript (as I think it shows in its final form) ensures that this work is based on a ‘model’.

__

Mat & Meth

In developing predictive models (simulators) to be applied in health, one needs to consider, first of all (among other variables) the benefits for the patient(s) which needs to be measured based on ‘physiological’ parameters. The M&M described in detail, which variables (n= 7, in this case) are of interest, and how they’ve been calculated (which is of the utmost relevance) for both the mechanical and the biological components. Notably, their model showed a correlation close to 1 (R=0.99, SE=0.02).

One of the most relevant aspects is that the authors considered, in their biological component, variables such as angiogenesis, ossification, chondrogenesis (formation of cartilage), calcification, etc., which are fundamental, as these occur in the ‘real’ physiological environment.

——

In their results, the authors describe the callus healing of both types of plates-techniques analysed (LP vs DCL) with their simulation model. Having stated previously, that the latter is the standard. From the clinical perspective, the variables compared between both models, are relevant, such as bone and cartilage concentration, blood perfusion, etc. Outstanding, is the fact that they present this comparison over space (1) and time (2). First, in a real scenario, fractures are not bi-dimensional, therefore a 3D model (as presented here) is more adequate. Second, a static screenshot of the changes might be also hard to analyse PROGRESSION/EVOLUTION of the healing process. Again, in a real patient, clinicians we need to evaluate how this improves in time. For this reason I found the model and the variables compared, very convenient.

Although described in human patients, this model can also be explored/applied in veterinary fractures which are physiologically and in various aspects, similar to those occurring in human patients. This might also be mentioned in their conclusions, should the authors agree on this. Moreover, to move on clinical trials (human), most studies are at first, tested on animal models (also known as clinical trial Phase 0)

Results:

Post operative analysis of the three main components in the ROI, (1) blood perfusion, (2) cartilage concentration and (3) bone concentration, revelaed minor differences along the healing period (8 weeks), no apparent difference between LP and DCP was observed by the end of week 8. The authors furhter support this by breaking down the distribution of the different types of tissue along the healing period with no difference between LC and DCP in the percentage of woven, cartilage and connective tissue. Moreover, theses results were further comapred with two other studies, one is a model and the other an animal experiment. Here the authors noted accurately, that the present study showed an IDENTICAL trend to that study of Lacroix, et al. 2002. In medicine and clinical trials, trends are more important than isolated values, since htey predict or help to estimate a prognosis or the clinical outcome (which we always have to tell to our patients or their relative).

Differences with the animal model were more apparent, however, the authors highlighted the role of the input parameters, in the bone healing progression. As I mentioned above, one of the main powers of these studies, is that irrespective of their caveats given that they are models, is that one the one hand are cheaper and quicker to perform, and more importantly, that they provide information towards the different components that might be involved in a 'real bilogical/medical/clinical' scenario, in which case, there is always other factors that cannot be measured. Importantly, the authors contempalted the possibility of a in vivo study in the future. With regard to the predictive value of this model and the 'trend' discused above, this point was raised (and corrected) by previous reviewers (Page no. 9, line: 22-25), here they clarify that their model is in good agreement with animal experiments.

I think that the second title (after concerns were raised by one reviewer) fits better with the study, HOWEVER, I would condition the approval of the manuscript for publication in PLos ONE as long as the word 'MODEL' is included in the NEW title. This is of the utmost relevance, precisely to avoid overinterpretation of their results. And to let the reader know, from the very begining that this study is based in a model (mostly) and not experimental data. This is important, since as it can be noted across the entire manuscript, the workd model is used abundantly.

I also ackowledge the value of the corrections and additional information that the authors amended after suggestions made by reviewers (or explain furhter when this was not possibly, as in the case of providing references given that no previous studies have been addressed the influence of screw type in fracture healing).

Some points were raised related to the use of passive/active voice, I agree with the aithors that in many aspects in scientific writing, we are encouraged to use passive voice. Other concerns of the meaning of language and jargon were raised too, but to me, the vocabulary was clear or when I had to, I searched for the meaning in either graphs or elsewhere and I could find it (at least in the revised form that I revised). Likewise, I noted that authors ammended and provided all the concerns from previous reviewers (data, clarity, references, etc), agreeing in most cases, which reflects their flexibility and scientific openess to discuss their work, which is also of great value. However, I had to agree with both previous reviewers that in some aspects the manuscript was not clear (maybe even less easy in its initial form) and that information of value was missed. Possibly some of this issues are down to the very technical terminology.

7. PLOS authors have the option to publish the peer review history of their article (what does this mean?). If published, this will include your full peer review and any attached files.

Reviewer #1: No

Reviewer #2: **Yes: **Hemant Unadkat

Reviewer #3: No

---

## [Author Response · Author response to Decision Letter 1]

19 May 2022

All comments are addressed and a point-to-point response file is attached.

---

## [Decision Letter · Decision Letter 2]

23 Jun 2022

A preclinical model of post-surgery secondary bone healing for subtrochanteric femoral fracture based on fuzzy interpretations

PONE-D-21-37870R2

Dear Dr. Chanda,

We’re pleased to inform you that your manuscript has been judged scientifically suitable for publication and will be formally accepted for publication once it meets all outstanding technical requirements.

Kind regards,

Yiming Tang, Ph.D.

Academic Editor

PLOS ONE

Additional Editor Comments (optional):

Reviewers' comments:

Reviewer's Responses to Questions

**Comments to the Author**

1. If the authors have adequately addressed your comments raised in a previous round of review and you feel that this manuscript is now acceptable for publication, you may indicate that here to bypass the “Comments to the Author” section, enter your conflict of interest statement in the “Confidential to Editor” section, and submit your "Accept" recommendation.

Reviewer #2: All comments have been addressed

Reviewer #3: All comments have been addressed

2. Is the manuscript technically sound, and do the data support the conclusions?

Reviewer #2: Yes

Reviewer #3: Yes

3. Has the statistical analysis been performed appropriately and rigorously? 

Reviewer #2: Yes

Reviewer #3: I Don't Know

4. Have the authors made all data underlying the findings in their manuscript fully available?

Reviewer #2: Yes

Reviewer #3: Yes

5. Is the manuscript presented in an intelligible fashion and written in standard English?

Reviewer #2: Yes

Reviewer #3: Yes

6. Review Comments to the Author

Reviewer #2: The manuscript is ready for publication in its present form. All the concerns from this reviewer have been suitably addressed.

Reviewer #3: I congratulate the authors for addressing positively all the reviewers comments (positive and negative ones).

7. PLOS authors have the option to publish the peer review history of their article (what does this mean?). If published, this will include your full peer review and any attached files.

Reviewer #2: No

Reviewer #3: No

---

## [Editor Report · Acceptance letter]

13 Jul 2022

PONE-D-21-37870R2 

A preclinical model of post-surgery secondary bone healing for subtrochanteric femoral fracture based on fuzzy interpretations 

Dear Dr. Chanda:

I'm pleased to inform you that your manuscript has been deemed suitable for publication in PLOS ONE. Congratulations! Your manuscript is now with our production department. 

Kind regards, 

on behalf of

Professor Yiming Tang 

Academic Editor

PLOS ONE